# Applicability of a Chemiluminescence Immunoassay to Screen Postmortem Bile Specimens and Its Agreement with Confirmation Analysis

**DOI:** 10.3390/ijms25073825

**Published:** 2024-03-29

**Authors:** Martina Franzin, Rachele Ruoso, Monica Concato, Davide Radaelli, Stefano D’Errico, Riccardo Addobbati

**Affiliations:** 1Institute for Maternal and Child Health, IRCCS “Burlo Garofolo”, Via dell’Istria 65/1, 34137 Trieste, Italy; martina.franzin@burlo.trieste.it (M.F.); rachele.ruoso@burlo.trieste.it (R.R.); 2Department of Medical, Surgical and Health Sciences, University of Trieste, 34149 Trieste, Italy; monica.concato@studenti.units.it (M.C.); davide_radaelli@hotmail.it (D.R.); sderrico@units.it (S.D.); 3Department of Forensic Medicine, Azienda Sanitaria Universitaria Giuliano Isontina, 34149 Trieste, Italy

**Keywords:** postmortem, bile, forensic, toxicology

## Abstract

Bile has emerged as an alternative matrix for toxicological investigation of drugs in suspected forensic cases of overdose in adults and intoxications in children. Toxicological investigation consists in screening and, subsequently, confirming the result with specific techniques, such as liquid chromatography with tandem mass spectrometry (LC-MS/MS). As there is no screening test on the market to test postmortem bile specimens, the novelty of this study was in investigating the applicability of a chemiluminescence immunoassay, designed for other matrices and available on the market, on bile and validate its use, testing the agreement with LC-MS/MS analysis. Bile specimens were obtained from 25 forensic cases of suspected death from overdose and intoxication. Sample preparation for bile screening consists simply in centrifugation and dilution. Confirmation analysis allows simultaneous identification of 108 drugs and was validated on bile. Kappa analysis assessed a perfect agreement (0.81–1) between the assays for benzodiazepines, methadone, opiates, cocaine, oxycodone, cannabinoids, buprenorphine and pregabalin; a substantial agreement (0.41–0.6) was reported for barbiturates. No agreement was assessed for amphetamines, due to an abundance of putrefactive amines in postmortem specimens. In conclusion, this fast and easy immunoassay could be used for initial screening of bile specimens, identifying presence of drugs, except amphetamines, with reliability.

## 1. Introduction

At present, death occurring from assumption of illicit drugs has increased, probably due to the diffusion of novel molecules and their greater availability [1]. In particular, over the past 10 years, the use of drugs, such as synthetic opioids, psychostimulants with abuse potential and cocaine, and consequent deaths have largely incremented; the most affected by this scenario are young subjects and adults with mental illness [1,2,3]. As well as overdose cases in adolescents and adults, pediatric intoxication has become not so unusual, due to the natural tendency of children to explore their environment: mostly children aged 1–4 were exposed to pharmaceuticals such as analgesics, psychotropics and cardiovascular drugs [4].

Death scene investigations and clinical information provided by autopsy are not sufficient to shed light on forensic cases; therefore, postmortem forensic toxicology has a fundamental role in determining the assumption of drugs of abuse, thus elucidating the cause of death [5].

In this context, the forensic toxicology laboratory develops, validates and applies methods for the determination and confirmation of illicit substances in postmortem specimens. Usually, the evaluation of the presence of drugs must be performed on at least two matrices because of the intrinsic sample variability due to after-death biochemical changes that modify the pre-existing environment [6,7].

Blood and urine are the traditional biological fluids handled by forensic toxicologists to determine drug presence at the time of the death or within a period before death, when drugs have already been metabolized, respectively [8,9]. Sometimes, mainly in case of severe decomposition, it is not possible to use these matrices because they are not available for removal, collection and analysis [8].

To overcome this limitation, non-conventional matrices, such as vitreous humor, hair, oral fluid and bile, have been introduced in forensic toxicology analysis [8,10,11]. In particular, bile has emerged as alternative biological fluid in case of unavailability of urine matrix [12]. Indeed, this biological fluid is indicative of past drug exposure, differently from other matrices that reflect blood concentration (for instance, vitreous humor) [8]. Furthermore, bile is more convenient to analyze and assess past drug use in comparison to other biological material, such as hair, because of the faster and easier sample preparation, as well as the reliability of results obtained, due to matrix homogeneity [12,13]. However, bile could be used complementary to traditional matrices, because it is easy to collect and large in volume [8,14]. Indeed, even 50 mL can be obtained through syringe aspiration from the gallbladder or incision compression if bile is too viscous [8,15]. Noteworthy, an advantageous peculiarity of bile consists in its extended detection window for measuring drugs, much wider than other matrices [16,17]. Indeed, after hepatic metabolism, drugs and their derivatives are excreted into bile and concentrated in the gallbladder before being eliminated [16,17].

The common approach for a forensic toxicological investigation comprises the screening of the postmortem sample, to identify one or more drugs of abuse, and, subsequently, the confirmation of drugs’ positivity [18,19]. In detail, screening tests consist mainly in immunoassays that allow rapid results and detection of a large amount of molecules belonging to a pharmacological class; in contrast, confirmation analyses use more specific techniques, such as liquid chromatography coupled with tandem mass spectrometry (LC-MS/MS), to confirm a target molecule and quantify it [19]. Generally, a high agreement between the screening tests and the confirmation analyses is desired.

Chips with different analytical reagents (antigens, antibodies, nucleic acid probes) arrayed on have been developed and put on the market; among these technologies, the Evidence MultiStat in combination with Biochip Array Technology by Randox uses a chemiluminescence reaction as output signal and allows one to perform several laboratory-based assays [20,21]. Interestingly, chemiluminescence immunoassays have been previously used on postmortem bile specimens to determine suspected fatal insulin administration, producing suitable data in accordance with other more specific techniques [22].

Unfortunately, the Randox Evidence MultiStat, similar to other tests on the market, has been developed and validated for the analysis of traditional biological fluids and there is the need to ascertain its efficacy on other non-conventional, but used, matrices, such as bile. Therefore, the present study investigated the applicability of this screening test to bile and evaluated its reliability, comparing the results obtained on specimens from 25 forensic cases with those from LC-MS/MS confirmation analysis, developed and validated on this specific biological matrix.

## 2. Results

### 2.1. Immunoassay Analysis

Postmortem bile specimens obtained from 25 forensic cases with suspected cause of death due to overdose or intoxication were screened and results, expressed qualitatively (positive or negative), are reported in Table 1. Eighteen (72%) resulted as positive to the screening tests. In detail, the immunoassay allowed the detection of 10/20 classes of illicit substances detectable in our cohort. Furthermore, co-assumption of drugs of abuse in most forensic cases was demonstrated by the present analyses.

### 2.2. Confirmation Analysis

Confirmation analysis was conducted on all postmortem bile specimens, previously screened, and qualitative results are reported in Table 2. Interestingly, all the negative results from the immunoassay were confirmed. Also, all the positive results regarding the classes of benzodiazepines, opiates, cocaine, oxycodone, buprenorphine and pregabalin were confirmed.

Since LC-MS/MS analysis allowed us to obtain quantitative results, we reported the average concentration of each illicit drug identified with an indication of the standard deviation (Table 3). Quantitative results were obtained by dilution and calculation based on the dilution factor, when sample concentration was higher than the upper limit of quantification.

### 2.3. Agreement

Screening test performance, described by the percentages of concordant results between the screening and confirmation analysis and the percentages of false positives and negatives derived from the screening, is reported in Figure 1.

Beyond calculating the percentages of concordance of the assays, Cohen’s kappa analysis was performed to demonstrate the degree of agreement between screening and confirmation analysis, considering the chance agreement (Table 4). Interestingly, according to the K score, there is perfect agreement (0.81–1) between the assays regarding the identification of benzodiazepines, methadone, opiates, cocaine, oxycodone, cannabinoids, buprenorphine and pregabalin. Moreover, substantial agreement (0.41–0.6) was proven for the determination of barbiturates. It is worth noting that no agreement between the screening and confirmation tests was evidenced, when amphetamines were analyzed in bile matrix.

## 3. Discussion

As, sometimes, traditional matrices are not available and drugs are excreted via biliary route, the use of bile as an alternative biological matrix to assess past drug use has been introduced for postmortem toxicological investigations [8,16]. Indeed, its role becomes fundamental, particularly in case of unavailability of urine matrix; furthermore, use of bile, complementary or not to traditional matrices, is convenient because it is easy to collect and large in volume, and it presents a detection window greatly wider than that of other materials [8,15,16,23]. However, there is no formal approval of the use of this biological fluid in forensic toxicology investigations. Common reported drawbacks of bile as a study matrix consist mainly in the lack of correlation between bile and blood concentrations and over- or underestimation of drug concentration, due to the postmortem redistribution phenomenon [15]. For this last-mentioned reason, as occurring for other matrices, the quality of the results could be affected by the period occurring from death to the moment of sample collection [15].

To date, immunoassays on the market allow the screening of classes of illicit drugs in blood, urine, oral fluid, hair and tissue [24]. In this context, there is the need to define an analytical workflow, regarding both screening tests and confirmation analysis, to handle bile. To the authors’ knowledge, only the case report by Tassoni and colleagues described analysis of drugs of abuse, particularly opioids, in bile, even if they focused their attention on the forensic case and without screening before confirmation [25].Therefore, we adapted a pre-existent protocol, designed by a manufacturer for traditional matrices, on bile and described its screening performance, comparing it with a confirmation analysis previously developed and validated in our laboratory [26].

Interestingly, several immunoassays on the market, including the one used in this work, rely on an innovative biochip array technology and employ antibodies immobilized in predefined regions on the biochip to make chemiluminescence reactions occur, leading to a final qualitative result [24]. Screening tests with this technique are easy, fast (not more than 25 min) and allow simultaneous drug identification. As for the other matrices, for whom the immunoassay was validated for, bile does not require time-consuming and demanding sample preparation, according to our study.

Interestingly, a chemiluminescence immunoassay for insulin determination in bile has been previously reported for demonstrating fatal administration; furthermore, the work by Palmiere and colleagues highlighted results of the immunoassay concordant with other more specific techniques [22].

Noteworthy, the confirmation analysis consists in a LC-MS/MS method previously developed and validated on bile matrix, allowing one to quantify drugs of abuse with accuracy [26]; therefore, the comparison between the screening test and the confirmation analysis has great reliability thanks to the employment of the same biological matrix to construct calibrators, dissimilar to a previous study (McLaughlin et al., 2019). Furthermore, our confirmation analysis employed a unique LC-MS/MS method and is able to assess the presence of a large number of drugs of abuse in a chromatographic run, differently from what was observed in previous studies [24,25,27].

Previous works testing the agreement between the results obtained from chemiluminescence immunoassays and confirmation analysis for forensic purposes have been published, but no one has focused on bile [24,28]. Interestingly, taking into consideration Cohen’s kappa analysis, which determines the agreement also on the basis of chance, perfect agreement (0.81–1) was reported for benzodiazepines, methadone, opiates, cocaine, oxycodone, cannabinoids, buprenorphine and pregabalin and substantial agreement (0.41–0.6) was reported for barbiturates, as already demonstrated with analogue instrumentations on different matrices [24]. When sanctuary non-concordant results were reported for these classes of drugs of abuse, it may be due to co-assumption of several illicit drugs and other substances similar to the target molecules. Indeed, it is well known that immunoassays are not “specific” and cross-reactions could occur, not only causing false positives but also false negatives, as for false-negative methamphetamine results caused by the presence of nor-2-chlorpromazine sulfoxide [29,30]. Furthermore, false negatives could be intuitively caused by the sensitivity of chemiluminescence immunoassays, generally lower than the technique used for the confirmation analysis [31].

Despite the fact that the proposed immunoassay could discriminate amphetamine from methamphetamine itself through specific antibodies, the data obtained showed that there is no agreement between the screening and confirmation analysis regarding the amphetamines group. There is growing evidence about the cross-reactivity of sympathomimetic amines, such as epinephrine and norepinephrine, usually administered for the treatment of allergic reactions, in chemiluminescence immunoassays [32]. Nonetheless, false-positive results for amphetamines could mainly be explained by the presence of putrefactive amines, such as putrescine, cadaverine, phenethylamine and tyramine, produced largely by saprogenic bacteria in moderately to heavily decomposed bodies [24,32,33]. Therefore, in order to discriminate true positivity to illicit drugs from presumptive ones, confirmation analyses that use specific and sensitive techniques must be performed after initial screening.

Interestingly, the differences underlined by Cohen’s kappa analysis were not evident, calculating simply the percentage of agreement, indicating that the chance, estimated on the basis of the number of cases and the frequency of positive and negative cases, is a fundamental variable to take into consideration.

## 4. Materials and Methods

### 4.1. Chemicals and Reagents

Phosphate-buffered saline was purchased by Sigma-Aldrich (Milan, Italy). Reagents for toxicological screening (Drugs of Abuse array blood (DoA) kit) were purchased from Randox laboratories (Crumlin, UK). Reagents for toxicological confirmation (MassTox Drugs of Abuse testing Mobile phases A, B and rinsing solution, MassTox Drugs of Abuse Analytical column, 6Plus1 Multilevel Urine Calibrator SET, MassCheck Drugs of Abuse testing urine, MassTox Drugs of Abuse testing Internal Standard, MassTox Drugs of Abuse testing Enzyme solution set, MassTox Drugs of Abuse testing Precipitation reagent and Dilution buffer) were purchased from Chromsystems Instruments & Chemicals GmbH (Munich, Germany).

### 4.2. Biological Samples

Postmortem specimens of bile (25) were obtained from forensic cases with suspected cause of death due to intoxication or overdose. Bile samples (3 mL) were collected during autopsy by syringe aspiration from the gallbladder in tubes with adjuvants, such as sodium fluoride and potassium oxalate, to prevent putrefactive mechanisms. According to the experience of our laboratory, sample storage at −20 °C in the above-mentioned tubes without freezing and thawing cycles is recommended, since it does not allow degradation of the illicit drugs examined for at least 1 year. After forensic pathologists of the University of Trieste and School of Forensic Medicine performed sampling during autopsy, samples were carried to the Advanced Translational Diagnostic Laboratory for analysis. The results obtained were included in the toxicology case reports, following the guidelines by the Procurator Fiscal. Moreover, the use of leftover samples for analytical validation was approved by IRCCS Burlo Garofolo (RC 56/22).

Details related to the forensic cases investigated are reported concisely in Table 5.

### 4.3. Immunoassay Protocol

Screening tests were performed using the Randox Evidence MultiStat (Randox laboratories, Crumlin, UK), an immunoanalyzer that enables simultaneous detection of illicit drugs in blood, urine and oral fluid. The immunoanalyzer provides qualitative data, indicating values greater and lower than the cut-off as positive and negative, respectively. The accuracy of the immunoassay was certified by testing two levels of quality controls provided by the manufacturer before the sample analysis. Table 6 reports the drugs detected, as well as the instrumental cut-off concentrations.

The sample preparation was adapted to the biological matrix of interest. In detail, postmortem bile samples were centrifuged for 10 min at 14,500 rpm and diluted 1:10 in phosphate-buffered saline. Subsequently, 100 µL of sample was added to 300 µL of dilution buffer and 200 µL of diluted sample and 200 µL of cut-off, belonging to the DoA kit, was put on the cartridge and loaded on the instrument for measurement of the chemiluminescence reaction.

### 4.4. Confirmation by LC-MS/MS

Confirmation analysis was performed at the Advanced Translational Diagnostic Laboratory and all the analyzed samples underwent previous screening tests by immunoassays.

For the confirmation of the identification of illicit drugs in bile, a LC-MS/MS method, previously developed and validated on the matrix of interest, was used [26]. In detail, the analytical method is capable of confirming the identification of 108 drugs of abuse and metabolites belonging to amphetamines, benzodiazepines, cocaine and metabolites, barbiturates and opioids in bile (Table 7). The response of each analyte was normalized on one of the corresponding internal standards (IS), consisting in the deuterated compound.

Bile specimens were diluted 1:100 in phosphate-buffered saline. Lyophilized calibrators and quality controls were reconstituted with the free-dug and diluted matrix of interest. To 50 µL of sample/calibrator/quality control, 10 µL of internal standard mix, composed by the corresponding deuterated molecules, and 40 µL of β-glucuronidase enzyme were added and samples were incubated for 2 h at 45 °C to allow enzymatic deconjugation. After incubation, 100 µL of precipitant reagent was added and, after vortexing, the sample was centrifuged for 5 min at 14,500 rpm. Subsequently, 150 µL of dilution buffer was added to 100 µL of supernatant and 10 µL was injected into the instrument.

An HPLC Exion LC 2.0 (Sciex, Milano, Italy) combined with a QTRAP 6500  +  system (Sciex, Milano, Italy) was used to perform LC-MS/MS confirmation analysis. Chromatographic separation was achieved by eluting mobile phases A and B (MassTox Drugs of Abuse, Chromsystems) on an analytical column (MassTox Drugs of Abuse, Chromsystems) at a flow rate of 0.4 mL/min with the following gradient: 0–0.2 min isocratic 0% B, 0.2–10.2 linear gradient 100% B, 10.2–12.0 isocratic 100% B, 12.0–12.1 linear gradient 0% B, and 12.1–14 isocratic 0% B.

After being introduced in the mass spectrometer, samples were ionized both positively and negatively via electrospray ionization. Mass spectrometer source-dependent parameters are reported in Table 8.

Multiple reaction monitoring (MRM) mode was adopted. In particular, 2 MRM transitions (quantifier and qualifier) for each analyte and 1 for IS were monitored. Compound-dependent parameters, such as declustering potential (DP), entrance potential (EP), collision energy (CE) and collision cell exit potential (CXP), were optimized via instrument tuning mode, injecting standards with known compositions (Appendix A).

The International Council for Harmonization guideline Q2(R2) on validation of analytical procedures was used for analytical validation. In particular, linearity was assessed by constructing calibration curves with an R^2^ > 0.99 in 3 different analytical runs. Appropriate accuracy (percentage of accuracy of 100  ±  15%) and reproducibility (coefficient of variation < 15%) were assessed, testing 3 levels of quality controls (QCI, QCII and QCIII) intra- (3 times during an analytical run) and inter-daily (3 different analytical run). As reproducibility was optimal, measurements of samples were performed once. Sensitivity was evaluated by diluting several times a standard with known concentration and determining the lower limits of quantification and of detection based on the accuracy of the concentration calculated and the signal detectable, respectively (Appendix A). Also, the matrix effect, recovery and efficiency of the entire process of preparation and analysis were assessed based on the Matuszewski method (Appendix A) [34].

### 4.5. Statistical Analysis

Data processing and analysis were performed using Analyst (version 1.7) and Multiquant (version 3.0.2) software. Concentration of illicit drugs was calculated by normalization of the response ratio of analytes with one of the internal standards and by interpolation with the calibration curve. Calibration curves were fit by linear regression with weighting by 1/χ^2^, without forcing the line through the origin.

Percentage of agreement between the results of the screening test and the confirmation analysis was calculated by dividing the cases confirmed on the totality of cases by each compound. Furthermore, Cohen’s kappa analysis was also performed as statistical measurement to observe the agreement between the data sets, also taking into account the chance agreement. In particular, Kappa scores between 0.81–1 represent perfect agreement, 0.61–0.8 substantial agreement, 0.41–0.6 moderate agreement and 0.1–0.2 slight agreement [35]. Negative values may generally be interpreted as no agreement [35].

## 5. Conclusions

We proposed a workflow for bile analysis using a fast and easy chemiluminescence immunoassay, designed for other biological matrices and available on the market, and described the performance of this screening test on several classes of drugs of abuse. Interestingly, there is agreement between the screening and confirmation analysis for all classes of illicit drugs, except for amphetamines, whose unsatisfactory results are by now well known and attributed to interferences by putrefactive amines in immunoassays. Therefore, the proposed immunoassay could be used for initial screening of postmortem bile specimens, identifying the presence of illicit drugs, except amphetamines, with reliability.

Limitations of the study comprise the impossibility of obtaining quantitative data from the screening test and the impossibility to perform analytical validation. Our study underscores the vital need to validate positive samples rigorously using advanced chromatography techniques. This validation step ensures the accuracy and reliability of our findings, significantly enhancing the robustness of forensic toxicology assessments. Forensic toxicologists have to understand the capability of on-site and fast drug testing devices but also must be aware of their limitations.

## Figures and Tables

**Figure 1 ijms-25-03825-f001:**
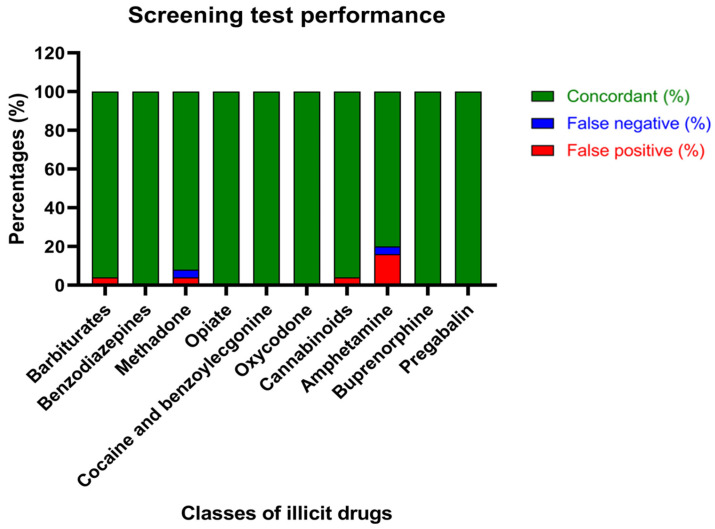
Screening test performance: percentages of concordant results between the screening and confirmation analysis (green), false negatives (blue) and false positives (red).

**Table 1 ijms-25-03825-t001:** Results of the screening tests on 25 postmortem bile specimens using Randox Evidence MultiStat with indication of number of positive/negative cases and percentage of frequency according to the screening test.

Classes of Illicit Drugs	Positive Cases (%)	Negative Cases (%)
Barbiturates	2 (8%)	23 (92%)
Benzodiazepines	11 (44%)	14 (56%)
Methadone	11 (44%)	14 (56%)
Opiate	8 (32%)	17 (68%)
Cocaine and benzoylecgonine	11 (44%)	14 (56%)
Oxycodone	1 (4%)	24 (96%)
Cannabinoids	6 (24%)	20 (76%)
Amphetamine	4 (16%)	21 (84%)
Buprenorphine	1 (4%)	24 (96%)
Pregabalin	2 (8%)	23 (92%)

**Table 2 ijms-25-03825-t002:** Results of the confirmation analysis on 25 postmortem bile specimens using LC-MS/MS with indication of number of positive and negative cases and percentage of frequency according to the confirmation analysis.

Classes of Illicit Drugs	Positive Cases (%)	Negative Cases (%)
Barbiturates	1 (4%)	24 (96%)
Benzodiazepines	11 (44%)	14 (56%)
Methadone	11 (44%)	14 (56%)
Opiate	8 (32%)	17 (68%)
Cocaine and benzoylecgonine	11 (44%)	14 (56%)
Oxycodone	1 (4%)	24 (96%)
Cannabinoids	5 (20%)	20 (80%)
Amphetamine	1 (4%)	24 (96%)
Buprenorphine	1 (4%)	24 (96%)
Pregabalin	2 (8%)	23 (92%)

**Table 3 ijms-25-03825-t003:** Quantitative results of LC-MS/MS analysis including the average concentration of illicit drugs identified, the standard deviation (SD) and the number of cases. n.a., not available.

Drugs Identified	Average Concentration (ng/mL)	SD	n. Cases
Amobarbital	2074.6	n.a.	1
Pentobarbital	2877.0	n.a.	1
Alprazolam	448.5	126.6	3
Hydroxy-Alprazolam	714.0	824.9	4
Clonazepam	741.0	940.9	3
Aminoclonazepam	5760.7	6038.2	4
Diazepam	127.0	40.6	3
Nordiazepam	330.0	65.0	4
Lorazepam	1443.6	2135.6	4
Lormetazepam	1076.9	77.6	2
Oxazepam	103.3	39.6	3
Temazepam	611.7	811.8	4
Methadone	5602.5	3610.9	10
EDDP	26,539.0	23,839.7	11
Codeine	53.9	40.0	4
Norcodeine	24.0	n.a.	1
Morphine	14,663.6	19,154.8	7
Cocaine	2747.5	3842.8	7
Cocaethylene	n.a.	n.a.	1
Norcocaine	149.4	181.8	7
benzoylecgonine	2998.3	4161.3	11
Oxycodone	187.9	n.a.	1
11-Nor-9-carboxy-Δ9-tetrahydrocannabinol	64,891.3	61,602.2	5
Amphetamine	111.0	n.a.	1
Buprenorphine	2.1	n.a.	1
Pregabalin	1320.9	320.9	2

**Table 4 ijms-25-03825-t004:** Percentage of false positives and false negatives of the screening test and percentage of agreement between the assays and indication of kappa score (K).

Classes of Illicit Drugs	False Positives of Screening Test (%)	False Negatives of Screening Test (%)	Agreement (%)	K
Barbiturates	4	0	96	0.65
Benzodiazepines	0	0	100	1
Methadone	3.9	3.9	92.2	0.84
Opiates	0	0	100	1
Cocaine and benzoylecgonine	0	0	100	1
Oxycodone	0	0	100	1
Cannabinoids	4	0	96	0.88
Amphetamines	16	4	80	−0.07
Buprenorphine	0	0	100	1
Pregabalin	0	0	100	1

**Table 5 ijms-25-03825-t005:** Characteristics of forensic cases. Categorical data are defined as numbers (percentage of the subgroup/total); numerical data are defined as medians (interquartile range (IQR)).

Forensic Case Characteristics	
Demographic data	
Age (years)	35 (28–51)
Male	69%
Medical history	
Reported drug abuse	36%
Family history of sudden cardiac death	4%
Steatohepatitis	4%
Medico-legal data	
Time between death and autopsy	5.5 (4–7.25)
Natural death	12%
Death from cardio-respiratory arrest	12%
Death from respiratory failure	16%
Sudden cardiac death	8%
Death from intake of caustic substances	4%
Death from stab wound	4%
Not documented	44%

**Table 6 ijms-25-03825-t006:** Drugs detected and cut-off concentrations of Randox Evidence MultiStat screening test.

Drugs Detected	Cut-Off Value (ng/mL)
Fentanyl	1
AB-PINACA	2
Methamphetamine	50
Barbiturates	50
Benzodiazepines	20
AB-CHMINACA	5
Methadone	10
Opiate	80
Phencyclidine	5
Cocaine and Benzoylecgonine	25
Oxycodone	10
Tramadol	5
Cannabinoids	10
Tricyclic antidepressant	60
Amphetamine	50
Buprenorphine	2
6-monoacetylmorphine	10
Alpha-PVP	5
Pregabalin	1000
Ethyl Glucuronide	500

**Table 7 ijms-25-03825-t007:** Illicit drugs identified by LC-MS/MS analysis and the IS used for the normalization.

Analytes	IS Used
Amphetamine	Amphetamine-D5
3,4-methylenedioxyphenyl-2-butanamine (BDB)	BDB-D2
Butylone	Butylone-D2
4-bromo-2,5-dimethoxyphenethylamine (2C-B)	2C-B-D5
4-iodo-2,5-dimethoxyphenethylamine (2C-I)	2C-I-D5
Cathinone	Cathinone-D5
3,4-methylbenzodioxolylbutanamine (MBDB)	MBDB-D4
3,4-methylenedioxyamphetamine (MDA)	MDA-D4
3,4-methylenedioxy-N-ethylamphetamine (MDEA)	MDEA-D4
3,4-metilenediossimetanfetamina (MDMA)	MDMA-D2
3,4-methylenedioxypyrovalerone (MDPV)	MDPV-D7
Mephedrone	Mephedrone-D3
Methamphetamine	Methamphetamine-D4
Methaqualone	Methaqualone-D6
Methylone	Methylone-D2
Methylphenidate	Methylphenidate-D10
para-methoxyamphetamine (PMA)	MDA-D4
Ritalinic acid	Ritalinic-acid-D4
Alprazolam	Alprazolam-D6
7-aminoclonazepam	7-NH2-Clonazepam-D4
7-aminoflunitrazepam	7-NH2-Flunitrazepam-D3
7-aminonitrazepam	7-NH2-Nitrazepam-D5
Bromazepam	Bromazepam-D4
Brotizolam	Diazepam-D4
Chlordiazepoxide	Chlordiazepoxide-D5
Clobazam	Clobazam-D8
Clonazepam	Clonazepam-D4
Demoxepam	Demoxepam-D5
Desalkylflurazepam (DA-Flurazepam)	DA-Flurazepam-D4
Desmethylflunitrazepam (DM-flunitrazepam)	DM-Flunitrazepam-D4
Diazepam	Diazepam-D4
Estazolam	Estazolam-D4
Flunitrazepam	Flunitrazepam-D3
Flurazepam	Flurazepam-D10
α-hydroxyalprazolam (OH-Alprazolam)	OH-Alprazolam-D5
α-hydroxymidazolam (OH-Midazolam)	OH-Midazolam-D4
α-hydroxytriazolam (OH-Triazolam)	OH-Triazolam-D6
3-hydroxybromazepam (OH-Bromazepam)	OH-Bromazepam-D4
Lorazepam	Lorazepam-D4
Lormetazepam	Lormetazepam-D8
Medazepam	Medazepam-D4
Midazolam	Midazolam-D6
Nitrazepam	Nitrazepam-D5
Norclobazam	Norclobazam-D6
Nordiazepam	Nordiazepam-D5
Oxazepam	Oxazepam-D5
Prazepam	Prazepam-D4
Temazepam	Temazepam-D6
Triazolam	Triazolam-D6
Gabapentin	Gabapentin-D4
Pregabalin	Pregebalin-D4
Promethazine	Promethazine-D5
Quetiapine	Quetiapine-D3
Benzoylecgonine	Benzoylecgonine-D3
Cocaethylene	Cocaethylene-D2
Cocaine	Cocaine-D2
Norcocaine	Norcocaine-D2
11-nor-9-Carboxy-Δ9-THC	THC-COOH-D3
Acetylcodeine	Acetylcodeine-D3
Buprenorphine	Buprenorphine-D4
Codeine	Codeine-D6
Dihydrocodeine	Dihydrocodeine-D6
2-ethylidene-1,5-dimethyl-3,3-diphenylpyrrolidine (EDDP)	EDDP-D3
Fentanyl	Fentanyl-D5
Hydrocodone	Hydrocodone-D6
Hydromorphone	Hydromorphone-D3
Meconin	Meconin-D3
Meperidine	Meperidine-D4
Methadone	Methadone-D8
6-Monoacetylmorphine (6-MAM)	6-MAM-D6
Morphine	Morphine-D3
Naloxone	Naloxone-D5
Naltrexone	Naltrexone-D3
Norbuprenorphine	Norbuprenorphine-D3
Norcodeine	Norcodeine-D3
Norfentanyl	Norfentanyl-D5
Normeperidine	Normeperidine-D4
Nortapentadol	Tapentadol-D3
Nortilidine	Nortilidine-D2
O-Desmethyltramadol (O-DM-Tramadol)	O-DM-Tramadol-D6
Oxycodone	Oxycodone-D6
Oxymorphone	Oxymorphone-D3
Papaverine	Papaverine-D3
Propoxyphene	Propoxyphene-D4
Sufentanil	Sufentanil-D4
Tapentadol	Tapentadol-D3
Thebaine	Thebaine-D3
Tilidine	Tilidine-D5
Tramadol	Tramadol-D3
Zaleplon	Zaleplon-D3
Zolpidem	Zolpidem-D5
Zopiclone	Zopiclone-D3
Allobarbital	Secobarbital-D5
Amobarbital	Butalbital-D5
Barbital	Secobarbital-D5
Butalbital	Secobarbital-D5
Hexobarbital	Secobarbital-D5
Pentobarbital	Pentobarbital-D5
Phenobarbital	Phenobarbital-D5
Secbutabarbital	Butalbital-D5
Secobarbital	Secobarbital-D5
Thiopental	Secobarbital-D5
Ketamine	Ketamine-D4
Lysergic acid diethylamide (LSD)	LSD-D3
Mescaline	Mescaline-D9
Norketamine	Norketamine-D3
2-Oxo-3-hydroxy-LSD	Ritalinic-acid-D4
Phencyclidine (PCP)	PCP-D5

**Table 8 ijms-25-03825-t008:** Mass spectrometer source-dependent parameters.

Parameters	Values
Curtain gas	40 psig
Collision gas	High
Ion spray voltage	4500 (positive mode) and −4500 (negative mode)
Capillary temperature	450 °C
Ion source gas	60 psig

## Data Availability

Data will be available upon reasonable request to the corresponding author.

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
