# Peer review of "Applicability of a Chemiluminescence Immunoassay to Screen Postmortem Bile Specimens and Its Agreement with Confirmation Analysis"

_ijms, 2024, doi:10.3390/ijms25073825_

Round 1

Reviewer 1 Report

Comments and Suggestions for Authors

Review paper (Applicability of a chemiluminescence immunoassay to screen 2

postmortem bile specimens and its agreement with confirmation analysis)

General comments

In this study, we delve into the applicability of chemiluminescence immunoassay (CLIA) for screening postmortem bile specimens. While bile is infrequently used for routine drug screening, it remains a common specimen studied in drug-related deaths. Like urine, bile is a waste fluid, and it contains higher concentrations of target compounds and their conjugates compared to blood. However, despite these advantages, using the bile to assess chronic drug use or determine the cause of deaths has not yet received formal approval.

One critical challenge lies in the poor correlation between bile drug concentrations and those in blood and other specimens. The inherent variability in bile composition among individuals further complicates its reliability as a matrix for drug detection. Additionally, bile contains various metabolites that may not directly correlate with actual drug exposure. Furthermore, the presence of putrefactive amines in postmortem specimens can interfere with accurate analysis.

Despite these hurdles, existing reagents primarily designed for whole blood testing are often sufficient for detecting drug presence in bile. Researchers must carefully justify their choice of bile testing techniques, especially considering that bile primarily serves as a screening test. Even if a sample tests positive, subsequent confirmation using LC-MS/MS remains essential.

1-Introduction:

The introduction is well-written and provides a clear context and motivation for the study.

-line 46-50 (Researchers must justify their choice of bile testing techniques, especially considering its primary role as a screening test. Vitreous humor, which aligns better with blood concentrations, could be a valuable alternative. The manuscript should provide clearer justifications for selecting bile as a matrix.)

2-Results

In order to enhance the clarity and accessibility of our findings for the scientific community, I recommend including correlation graphs that illustrate the relationships between the various techniques used. Graphical representations not only enhance the visual appeal but also facilitate easier comprehension of the data.

3-Disucssions

I recommend omitting any sentences related to Randox innovation and instead emphasizing the scientific

The authors should address false positives, false negatives, true positives, and true negatives in their investigation. Additionally, they must explore why the technique is less selective for the amphetamine group. Recognizing that this method serves as a primary test, they should carefully discuss its limitations. Any positive samples should undergo confirmation using advanced chromatography techniques.

2. Materials and Methods

" In the ‘Materials and Methods’ section, we emphasize the importance of comprehensive method validation for our study focusing on multidrug screening in bile specimens using immunoassay. Specifically, we recommend providing detailed validation information for the Randox Evidence Multistat (from Randox Laboratories, Crumlin, England). This validation should cover essential parameters such as linearity, sensitivity, selectivity, and precision.

Additionally, the manuscript would benefit from more extensive details regarding the LC-MS/MS method validation. We suggest including specifics about the instrument used, its complete settings, temperature conditions, program parameters, and column characteristics. To enhance clarity and facilitate understanding, consider presenting this critical information in a tabular format.

Furthermore, it is essential to augment the manuscript by providing additional details related to the cases investigated. A concise table summarizing relevant information, including age, gender, medical history, other detected drugs, and cause and manner of deaths, would enhance the overall scientific rigor of our work.

3-Results

The results section is quite lengthy, with most of the findings presented in tables, figures, and accompanying text. To enhance readability and follow the content more easily, I recommend summarizing this section.

In Table 5, the authors present cutoff values for the investigated drugs. These values are based on their judgment, considering that no bile screening test has been conducted.

 5- Conclusions

Revised Conclusion and I suggest this to be added (our study underscores the vital need to validate positive samples rigorously using advanced chromatography techniques. This validation step ensures the accuracy and reliability of our findings, significantly enhancing the robustness of forensic toxicology assessments.)

Author Response

General comments

In this study, we delve into the applicability of chemiluminescence immunoassay (CLIA) for screening postmortem bile specimens. While bile is infrequently used for routine drug screening, it remains a common specimen studied in drug-related deaths. Like urine, bile is a waste fluid, and it contains higher concentrations of target compounds and their conjugates compared to blood. However, despite these advantages, using the bile to assess chronic drug use or determine the cause of deaths has not yet received formal approval.

One critical challenge lies in the poor correlation between bile drug concentrations and those in blood and other specimens. The inherent variability in bile composition among individuals further complicates its reliability as a matrix for drug detection. Additionally, bile contains various metabolites that may not directly correlate with actual drug exposure. Furthermore, the presence of putrefactive amines in postmortem specimens can interfere with accurate analysis.

Despite these hurdles, existing reagents primarily designed for whole blood testing are often sufficient for detecting drug presence in bile. Researchers must carefully justify their choice of bile testing techniques, especially considering that bile primarily serves as a screening test. Even if a sample tests positive, subsequent confirmation using LC-MS/MS remains essential.

Answer: We would like to thank the reviewer for taking the time to review this manuscript. Please find the detailed responses below and the corresponding revisions in track changes in the re-submitted files.

1-Introduction:

The introduction is well-written and provides a clear context and motivation for the study.

-line 46-50 (Researchers must justify their choice of bile testing techniques, especially considering its primary role as a screening test. Vitreous humor, which aligns better with blood concentrations, could be a valuable alternative. The manuscript should provide clearer justifications for selecting bile as a matrix.)

Answer: We would like to thank the reviewer for the suggestions. We ameliorated the introduction paragraph justifying deeply the use of bile as alternative to urine matrix indicative of past drug exposure (lines 57-72) comparing bile with other used matrices. Furthermore, we mentioned a previous use of chemiluminescence immunoassay on bile postmortem specimens producing suitable results (lines 85-87). Also, we highlighted the disadvantages of using this biological matrix in the discussion paragraph (lines 172-178).

2-Results

In order to enhance the clarity and accessibility of our findings for the scientific community, I recommend including correlation graphs that illustrate the relationships between the various techniques used. Graphical representations not only enhance the visual appeal but also facilitate easier comprehension of the data. 

Answer: We would like to thank the reviewer for the suggestion and we inserted Figure 1 in the results paragraph. In detail, the graph describes the screening test performance, indicating the percentages of concordant results between the assay, the false positive and false negative of the screening test. We agreed with the reviewer that the figure ameliorates the manuscript allowing immediate comprehension.

3-Discussions

I recommend omitting any sentences related to Randox innovation and instead emphasizing the scientific

Answer: We would like to thank the reviewer for the suggestion and we revised the discussion paragraph omitting the Randox innovation and emphasizing the fast and easy bile screening through a chemiluminescence immunoassay (lines 191-203).

The authors should address false positives, false negatives, true positives, and true negatives in their investigation. Additionally, they must explore why the technique is less selective for the amphetamine group. Recognizing that this method serves as a primary test, they should carefully discuss its limitations. Any positive samples should undergo confirmation using advanced chromatography techniques.

Answer: We would like to thank the reviewer for the comment and we revised the discussion paragraph accordingly. Please note that we addressed the reason of occurring false positive and false negative (lines 219-226), especially focusing on false positive for amphetamine group (lines 227-235). Lastly, we pointed that confirmation analysis is mandatory in forensic toxicology (lines 235-238).

  1. Materials and Methods

In the ‘Materials and Methods’ section, we emphasize the importance of comprehensive method validation for our study focusing on multidrug screening in bile specimens using immunoassay. Specifically, we recommend providing detailed validation information for the Randox Evidence Multistat (from Randox Laboratories, Crumlin, England). This validation should cover essential parameters such as linearity, sensitivity, selectivity, and precision.

Answer: We would like to thank the reviewer for the comment. Even if Randox Evidence Multistat allows multi-drug screening, the result provided from the screening test is qualitative (greater or lower than the instrumental cut-off value). As the unavailability of the quantitative data, the accuracy of the immunoassay could be only certified by testing the quality controls provided by the manufacturer (lines 280-283). Therefore, we decided to present this argument as a limitation in the conclusion paragraph (lines 371-372).

Additionally, the manuscript would benefit from more extensive details regarding the LC-MS/MS method validation. We suggest including specifics about the instrument used, its complete settings, temperature conditions, program parameters, and column characteristics. To enhance clarity and facilitate understanding, consider presenting this critical information in a tabular format.

Answer: We would like to thank the reviewer for the suggestions and we revised the manuscript accordingly. In detail, we inserted extensive details regarding the LC-MS/MS method, as well the validation process, in the revised manuscript (paragraph “Confirmation by LC-MS/MS”) and in the Supplementary material provided.  

Furthermore, it is essential to augment the manuscript by providing additional details related to the cases investigated. A concise table summarizing relevant information, including age, gender, medical history, other detected drugs, and cause and manner of deaths, would enhance the overall scientific rigor of our work.

Answer: We would like to thank the reviewer for the suggestions and we revised the manuscript inserting Table 5, reporting the forensic cases details.

3-Results

The results section is quite lengthy, with most of the findings presented in tables, figures, and accompanying text. To enhance readability and follow the content more easily, I recommend summarizing this section.

Answer: We would like to thank the reviewer for the suggestion and we revised all the results paragraph accordingly. In detail, we revised the manuscript deleting text, where tables were already explanatory.

In Table 5, the authors present cutoff values for the investigated drugs. These values are based on their judgment, considering that no bile screening test has been conducted.

Answer: We would like to thank the reviewer for the comment. We specified in the revised manuscript that the values presented refers to instrumental cut-off (line 284).

5- Conclusions

Revised Conclusion and I suggest this to be added (our study underscores the vital need to validate positive samples rigorously using advanced chromatography techniques. This validation step ensures the accuracy and reliability of our findings, significantly enhancing the robustness of forensic toxicology assessments.)

Answer: We would like to thank the reviewer for the suggestions and we revised the conclusion accordingly.

Reviewer 2 Report

Comments and Suggestions for Authors

The manuscript presented by Franzin and co-workers, entitled Applicability of a chemiluminescence immunoassay to screen postmortem bile specimens and its agreement with confirmation analysis”, is an article showing that the proposed immunoassay may be used for initial screening of bile postmortem specimens, identifying the presence of illicit drugs, except amphetamines, with reliability.

Although the work is interesting, it requires significant corrections before it is published. 

- abstract slightly presents the importance of this work. The novelty of the presented methodology should be exposed.  

Introduction, page 2, lines 53-54 - Sometimes, mainly in case of severe decomposition, it is not possible to use these matrices because they are not available for removal, collection and analysis [8]. – the disadvantages of the presented method and application of bile in the forensic analysis should be discussed in the manuscript. The questions are as follows:

- How to collect a bile sample at a crime scene?

- how the period from death to the moment of sample collection affects the quality of the results obtained?

- what is the optimal way to collect a bile sample?

- under what conditions the pieces should be stored?

- Is there a relationship between the sample storage time and the type of data obtained?

- Confirmation by LC-MS/MS (page 7) – the details of LC-MS/MS analysis should be presented (MS device parameters, ion source parameters).

- what chromatographic column was used? How was the chromatographic separation optimized? What is eluent A? What is eluent B?

- How was the MRM method optimized?

- what is the level of detection in the LC-MS/MS technique,

- how many times were the measurements repeated

- what is the reproducibility of the method

- what are the matrix effects?

- The part of the work covers the LC-MS/MS technique used in the authors' research, in my opinion, it is described too briefly.

Check and correct the English.

Check and correct the reference style.

Comments on the Quality of English Language

Minor English editing is needed.

Author Response

The manuscript presented by Franzin and co-workers, entitled “Applicability of a chemiluminescence immunoassay to screen postmortem bile specimens and its agreement with confirmation analysis”, is an article showing that the proposed immunoassay may be used for initial screening of bile postmortem specimens, identifying the presence of illicit drugs, except amphetamines, with reliability. Although the work is interesting, it requires significant corrections before it is published. 

Answer: We would like to thank the reviewer for taking the time to review this manuscript. Please find the detailed responses below and the corresponding revisions in track changes in the re-submitted files.

- abstract slightly presents the importance of this work. The novelty of the presented methodology should be exposed.  

Answer: We would like to thank the reviewer for the suggestion and we revised the abstract evidencing better the novelty of the present study (lines 18-21, line 29). In detail, the novelty of the study consists in investigating the applicability of a fast and easy chemiluminescence immunoassay, designed for other matrices and available on the market, on bile and validate its use testing the agreement with the LC-MS/MS confirmation analysis, leading to perform screening also on this unconventional matrix.

Introduction, page 2, lines 53-54 - Sometimes, mainly in case of severe decomposition, it is not possible to use these matrices because they are not available for removal, collection and analysis [8]. – the disadvantages of the presented method and application of bile in the forensic analysis should be discussed in the manuscript.

Answer: We would like to thank the reviewer for the suggestions and we implemented in the discussion describing the disadvantages of testing bile (lines 172-178). Also, we presented some limitations of the method in the conclusion paragraph (lines 371-377).

The questions are as follows:

- How to collect a bile sample at a crime scene?

Answer: We would like to thank the reviewer for the comment and we specified in the material and methods paragraph (lines 257, lines 262-263) that bile was collected during autopsy and not at crime scene.

- how the period from death to the moment of sample collection affects the quality of the results obtained?

Answer: We would like to thank the reviewer for the comment and we specified in the discussion paragraph that, also occurring for other matrices, the quality of results could be affected by the period from death to the moment of sample collection (lines 176-178).

- what is the optimal way to collect a bile sample?

Answer: We would like to thank the reviewer for the comment and we specified in the introduction paragraph (lines 67-68) the optimal way to collect bile. Therefore, we specified in the material and methods paragraph also the way of bile collection of our study (lines 257-259).

- Is there a relationship between the sample storage time and the type of data obtained?

Answer: We would like to thank the reviewer for the comment and we specified in the material and methods paragraph (lines 259-262) that previous and internal of our laboratory tests on bile samples were performed assessing the same results after 1 year of storage.

- Confirmation by LC-MS/MS (page 7) – the details of LC-MS/MS analysis should be presented (MS device parameters, ion source parameters).

- what chromatographic column was used? How was the chromatographic separation optimized? What is eluent A? What is eluent B?

- How was the MRM method optimized?

- what is the level of detection in the LC-MS/MS technique,

- how many times were the measurements repeated

- what is the reproducibility of the method

- what are the matrix effects?

The part of the work covers the LC-MS/MS technique used in the authors' research, in my opinion, it is described too briefly.

Answer: We would like to thank the reviewer for the suggestions and we implemented the materials and methods paragraph regarding the parameters of the LC-MS/MS analysis. Please find all the parameters of the method and of the validation process in the revised manuscript (paragraph “Confirmation by LC-MS/MS”) and in the Supplementary material provided.

Check and correct the English.

Check and correct the reference style.

Answer: We would like to thank the reviewer and we checked and corrected the English, as well as the reference style.

Round 2

Reviewer 1 Report

Comments and Suggestions for Authors

Accepted 

Reviewer 2 Report

Comments and Suggestions for Authors

The revised version of the manuscript presented by Franzin and co-workers meets my requirements. The Authors presented comments and answers to all of my questions and doubts. 

Comments on the Quality of English Language

Minor English editing is needed.